# Process Optimization, Amino Acid Composition, and Antioxidant Activities of Protein and Polypeptide Extracted from Waste Beer Yeast

**DOI:** 10.3390/molecules27206825

**Published:** 2022-10-12

**Authors:** Lisha Zhu, Jianfeng Wang, Yincheng Feng, Hua Yin, Huafa Lai, Ruoshi Xiao, Sijia He, Zhaoxia Yang, Yi He

**Affiliations:** 1State Key Laboratory of Biological Fermentation Engineering of Beer, Qingdao 266100, China; 2National R&D Center for Se-Rich Agricultural Products Processing, Hubei Engineering Research Center for Deep Processing of Green Se-Rich Agricultural Products, School of Modern Industry for Selenium Science and Engineering, Wuhan Polytechnic University, Wuhan 430023, China; 3Key Laboratory for Deep Processing of Major Grain and Oil, Ministry of Education, Hubei Key Laboratory for Processing and Transformation of Agricultural Products, School of Food Science and Engineering, Wuhan Polytechnic University, Wuhan 430023, China

**Keywords:** waste beer yeast, process optimization, enzymatic hydrolysis, amino acid composition, amino acid score, antioxidant activity

## Abstract

Repurposing of waste beer yeast (WBY) that a main by-product of brewing industry has attracted considerable attention in recent years. In this study, the protein and polypeptide were extracted by ultrasonic-assisted extraction and enzymatic hydrolysis with process optimization, which resulted in a maximum yield of 73.94% and 61.24%, respectively. Both protein and polypeptide of WBY were composed of 17 Amino acids (AA) that included seven essential amino acids (EAA), and typically rich in glutamic acid (Glu) (6.46% and 6.13%) and glycine (Gly) (5.26% and 6.02%). AA score (AAS) revealed that the threonine (Thr) and SAA (methionine + cysteine) were the limiting AA of WBY protein and polypeptide. Furthermore, the antioxidant activities of WBY polypeptide that lower than 10 kDa against hydroxyl radical, DPPH radical, and ABTS radical were 95.10%, 98.37%, and 69.41%, respectively, which was significantly higher than that of WBY protein (25–50 kDa). Therefore, the protein and polypeptide extracted from WBY can be a source of high-quality AA applying in food and feed industry. Due to small molecular weight, abundant AA, and great antioxidant activity, WBY polypeptide can be promisingly used as functional additives in the pharmaceutical and healthcare industry.

## 1. Introduction

Pollution growth, population explosion, and resource shortage prompt human beings to look for high-quality and readily available sources of protein. In fact, plants and animals are good resources of edible protein but still have the disadvantages of requiring plenty of land, water, and energy [1]. Conversely, obtaining protein by microorganism has the advantages of time-saving, low-cost, and easy-cultivate. The residues and wastes of agriculture, industry, and food are natural substrates for cultivating microorganism to reap abundant edible protein. Sen et al. reported that cultivating microalga *Chlorella* sp. in brewery wastewater (BWW) can not only greatly repair the quality of waste water but also significantly increase the accumulation of protein [2]. Moreover, some reports claimed that replacing meat proteins with microbial proteins contributes to address global warming and protect the forests [3,4]. 

Beer yeast, a kind of *Saccharomyces cerevisiae* (*S. cerevisiae*), traditionally used in the foods production of cheese, wine, bread and beer because of its great fermentation ability [5]. *S. cerevisiae* has a status of “Generally Recognized as Safe” for fermentation industry [1] and plays a significant role in the brewing process of beer for increasing peculiar flavor and nutrients [6]. In recent five years, the annual beer production of China is about 35.624 million kiloliters, which means beer drinks are popular with consumers. However, 10–15 tons of WBY pastes or residues were inevitably produced when 10^4^ hL of beer was made, which is a challenging problem for beer industry to make full use of WBY efficiently [7]. In fact, WBY has attracted considerable attention based on the evidence that it contains abundant proteins, amino acids, vitamins, carbohydrates and minerals. Best to our knowledge, WBY and its extraction are mainly used in five fields, such as animal feed, enzyme source, functional food applications, fermentation substrate and non-food applications [8]. On the one hand, a large proportion of WBY is used to produce animal feed [9], enhance flavor [10], improve texture [11,12] in food industry; on the other hand, WBY has the potential to synthesize catalyst for biodiesel production [13], bio-adsorb pigments [14] and metal ions [15,16]. Vieira reported that the 64% protein with considerable essential amino acids was extracted from lyophilized WBY [10], and the protein along with its hydrolysate can be regarded as a good protein supplement to apply in livestock feed [17,18] and microbial culture medium [19]. Lots of studies indicated that yeast protein is a good option to provide nutrition for human beings and animals along with cheap and environment friendly [1,10,17]. Furthermore, peptides extracted from WBY exhibited great bioactivities such as antioxidant and antihypertensive [20]. 

Current studies about extracting protein and polypeptide from WBY are listed in Table 1. So far, there are various ways to obtain WBY protein, for example, the yeast cell wall can be destroyed and removed by mechanical methods and enzymes hydrolysis [21]. Next, the peptide bonds in protein are broke by enzymatic hydrolysis so that small polypeptides as well as free AA are released. Among this, ultrasonic-assisted extraction, a green technic with saving time and energy, is widely employed to extract natural products and components [22]. This study was attached great attention on the optimal extraction of WBY protein and polypeptide with low-cost, energy-saved and efficient. Ultrasonic-assisted extraction and enzymatic hydrolysis were employed to obtain WBY protein and polypeptide, respectively, with process optimization, the molecular weight, AA composition, and antioxidant activity of the products were evaluated systematically. The results can contribute to the reuse of WBY and expand the utilization of WBY protein and polypeptide as functional ingredient in food industry.

## 2. Results and Discussion

### 2.1. Process Optimization of Extracting Protein

#### 2.1.1. The Content of Crude Protein in WBY

The results of Kjeldahl determination showed that the total crude protein of WBY was 41.928%, which was lined with the view that the protein account for 35% to 69% of dry yeast [24]. 

#### 2.1.2. Single-Factor Experiments

Figure 1 displays the results of single-factor experiments for extracting WBY protein with ultrasonic-assisted extraction. As shown in Figure 1, the extraction rate of WBY protein reached the maximum when the power of ultrasound, pH of extraction, and the solid–liquid ratio reached 300 W, 8.5, and 12%, respectively. The extraction rate rapidly went up from 48.87% to 66.01% as the ultrasound power increased from 250 W to 300 W. However, the extraction rate sharply dropped up to 36.40% with the ultrasound power continued to increase to 450 W (Figure 1B). Moreover, the change of extraction rate was not significant when the pH of extraction increased from 5.5 to 7.5. After that, the rate dramatically rose from 39.21% to 76.65% when the pH increased to 8.5, and then fell to 52.71% with the environment became more alkaline (Figure 1C). Finally, the extraction rate continued to grow steadily from 27.11% to 82.32% when the solid-liquid ratio between 6% and 12% but dropped to 70.18% subsequently as the ratio of solid-liquid exceeded 12% (Figure 1D). 

#### 2.1.3. RSM Experiments

Based on the results of the single-factor test, the following conditions were selected to further maximize the yield: a power range of 250–350 W, a pH range of 7.5–9.5, and a solid-liquid ratio range of 10–14% (Table 2). The variance analysis of the abovementioned regression model is shown in Table 3, wherein ANOVA reveals the significance of each coefficient. The coefficient of determination (R^2^) in the model was 0.9653, *p* value of model < 0.01, and the lack of fit term was not significant, indicating that the model was reasonable and available for predicting WBY protein extraction. The power of ultrasound (X_1_), pH of extraction (X_2_), solid-liquid ratio (X_3_) and three quadratic terms (X_1_^2^, X_2_^2^, and X_3_^2^) were statistically significant (*p* < 0.01), indicating that the factors of X_1_, X_2_ and X_3_ had a significant impact on extraction rate of WBY protein. Moreover, the influence of pH was much greater than that of power and solid-liquid ratio, revealing that the pH of extraction was the most important factor in WBY protein extraction using ultrasonic-assisted extraction. The linear regression equation of the experiment was obtained as follows:
Y = 73.36 + 4.79X_1_ + 4.98X_2_ + 3.64X_3_ + 4.17X_1_X_2_+ 4.15X_1_X_3_ + 1.77X_2_X_3_ − 8.51X_1_^2^ − 9.18X_2_^2^ − 6.81X_3_^2^

The results of 3D response surface and response contour plots of the three factors are shown in Figure 2. Figure 2A,B present the relationship between power of ultrasound and pH of extraction during WBY protein extraction. The extraction rate approached the peak (75.12%) when the two factors increased to approximately 317 W and 8.86, respectively. Thereafter, the extraction rate went down with a constant increase of either power or pH. Furthermore, the response surface of power and pH was the steepest compared with others, indicating that power and pH have the maximum impact on WBY protein rxtraction. Figure 2C,D depict the influence of power and solid-liquid ratio on extraction rate, the rate climbed to the maximum (74.95%) when the power and solid-liquid ratio were approached 318 W and 12.75%, respectively. The interaction effect of power and solid-liquid ratio was slightly smaller than that of power and pH. As shown in Figure 2E,F, the extraction rate increased from about 50.79% to 73.68% with pH and solid-liquid ratio added from 7.5, 10.0% to 8.5, 11.98%. Considering the operability of extraction, the solution of power, pH, and solid-liquid ratio that provided by Design-Expert 11 was simplified to 325 W, 8.9 and 13%, respectively. Proof tests were carried out using the abovementioned optimal parameters, and the WBY protein yield reached 73.94 ± 0.17%, which was close to the predicted yield of 76.5% (*p* < 0.01). Overall, the extraction rate of WBY protein was significantly influenced by pH of extraction and power of ultrasound. Alkalinity environment contribute to the interaction between protein and water molecules so that the solubility of most protein is increasing [27]. The purity of protein was 68.31 ± 0.24%, and most of the impurities are probably carbohydrates that derive from yeast cell wall with higher molecular weight [24]. Tanguler et al. reported that spent brewer’s yeast cells were induced to autolysis under 45 °C for 72 h, which yielded 76.4% yeast protein [25]. Lamoolphak et al. reported that 78% yeast cells were decomposed in water at 250 °C for 20 min, along with produced 0.16 mg protein from 1 mg dry yeast [23]. This study revealed that ultrasonic-assisted extraction is feasible for extracting protein from WBY with time-saving and efficient, along with protein yield of 73.94 ± 0.17%.

### 2.2. Process Optimization of Preparing Polypeptide

#### 2.2.1. Comparison of Two Types of Enzymes

Figure 3A manifests that there was no significant difference between trypsin and alkaline protease on WBY polypeptide yield within ten h. Afterwards, contrary to alkaline protease, WBY protein was continued to be hydrolyzed by trypsin between 10 and 14 h and the extraction rate of WBY polypeptide kept rising. The hydrolytic effect of trypsin was almost 1.25-fold that of alkaline protease after reacted for 24 h, with 50.88% and 40.35%, respectively. Thus, trypsin was selected for further study considering the extraction rate of WBY polypeptide.

#### 2.2.2. Single-Factor Experiments

The influence of environment and enzyme concentration on hydrolyze WBY polypeptide is shown in Figure 3. The extraction rate of WBY polypeptide went up when temperature ranged from 30 °C to 40 °C and then went down with the increasing of temperature, the maximum extraction rate was achieved at about 40 °C, as shown in Figure 3B. The yield of WBY polypeptide at 50 °C was about 66% of that at 40 °C, indicating that the optimal temperature of trypsin was approximately 40 °C and the activity of trypsin was reduced at excessive temperature. Moreover, the extraction rate% stayed above 40% when the pH of extraction varied from 7 to 8, while the activity of trypsin became weak in a more alkaline environment (Figure 3C). The extraction rate rapidly increased from 44.54% to 50.50% as the concentration of trypsin increased from 0.5% to 1.0%, but the yield of WBY polypeptide was changed little when the concentration of the trypsin added to 2.5% (Figure 3D). Overall, the extraction rate of WBY polypeptide that hydrolyzed by trypsin reached the maximum when temperature, pH, and enzyme concentration reached 40 °C, 8, and 1%, respectively. 

#### 2.2.3. RSM Experiments

The three factors were selected to further maximize the polypeptide yield: a pH range of 7–8, a temperature range of 35–45 °C, and an enzyme concentration range of 0.5–1.5% (Table 4) and the results of variance analysis are shown in Table 5. The coefficient of determination (R^2^) in the model was 0.9929, *p* value of model < 0.01, and the lack of fit term was not significant, indicating that the model was reasonable and available for predicting WBY polypeptide extraction. The temperature of ultrasound (X_2_), and three quadratic terms (X_1_^2^, X_2_^2^, and X_3_^2^) were greatly significant (*p* < 0.01), indicating that the temperature of environment had a significant impact on extraction rate of WBY polypeptide. The linear regression equation for predicting the extraction rate of WBY polypeptide was obtained as follows:
Y = 62.02+ 0.4375X_1_ + 1.05X_2_ + 0.395X_3_ − 0.715X_1_X_2_ + 0.09X_1_X_3_ − 0.37X_2_X_3_ − 2.52X_1_^2^ − 5.43X_2_^2^ − 3.87X_3_^2^

The results of 3D response surface and response contour plots of the three factors are shown in Figure 4. Figure 4A,B show the interaction between pH of extraction and temperature when extracting WBY polypeptide. The response surface of above factors was the steepest compared with others, suggesting that pH and temperature have the maximum impact on extracting WBY polypeptide. The extraction rate reached the peak (61.64%) when pH of extraction and temperature increased to 7.7 and 41.6 °C, respectively. Figure 4C,D depict the influence of pH and enzyme concentration on extraction rate, the rate climbed to the top (61.5%) when the two factors were reached 7.68 W and 1.18%, respectively. After that, the rate continued to go down with a constant increase of either pH or enzyme concentration. As shown in Figure 4E,F, the extraction rate increased from about 50.99% to 61.73% with the temperature and enzyme concentration increased from 35 °C, 0.5% to 41.3 °C, 1.13%. Considering the operability of extraction, the solution of pH, temperature and enzyme concentration that provided by Design-Expert 11 was simplified to 7.5, 41 °C and 1.0%, respectively. Proof tests were carried out using the abovementioned optimal parameters, and the WBY protein yield reached 61.24 ± 0.15%, which was close to the predicted yield of 62.09% (*p* < 0.01). Overall, the extraction rate of WBY polypeptide was greatly influenced by pH and temperature that directly affect the activity of trypsin, which lined with the report of Martin [21].

The DH and purity of WBY polypeptide were 42.76 ± 1.43% and 64.96 ± 0.63%, respectively, under the optimal condition. Manuela et al. reported that the autolysis products of spent brewer’s yeast were hydrolyzed by protease and filtrated by selective membrane with 3 kDa cut-off. The purity of hydrolysates varied from 30% to 69%, and 27–48% simple sugars of lower molecular weight existed in products [24]. Thus, hydrolysis by trypsin combined with membrane ultrafiltration is available for extracting WBY polypeptide, and the purity can be improved by removing carbohydrates. 

### 2.3. Molecular Weight Distribution

The SDS-PAGE gel electrophoresis of protein and polypeptide of WBY is shown in Figure 5A. It is clearly observed that most bands of WBY protein were located at 25–50 kDa, which implied that it is feasible to extract WBY protein using ultrasonic-assisted extraction under slightly alkaline environment. However, none of WBY polypeptide bands was observed in gel electrophoresis, which indicated that the molecular weight of polypeptide hydrolyzed by trypsin was smaller than 10 kDa. Lisboa et al. reported that the polypeptides obtained by hydrolyzing *Spirulina* sp. LEB 18 using alkaline protease exhibited smaller molecular weight with lower than 20 kDa, suggesting that the treatment of enzymatic hydrolysis greatly enhanced the solubility and absorption of protein [27]. Furthermore, Maria et al. reported that the polypeptides of lower than 3 kDa were obtained by hydrolyzing whey protein with 1.7% (*v*/*v*) trypsin at 41.1 °C for 4.31 h, which had strong antioxidant activity [28].

### 2.4. Amino Acid Composition

The amino acid composition of protein and polypeptide of WBY is shown in Table 6. The protein and polypeptide of WBY were composed of 17 amino acids, which included seven EAA and ten NEAA. They were rich in glutamic acid (Glu) (6.46% and 6.13%), glycine (Gly) (5.26% and 6.02%), lysine (Lys) (3.92% and 3.21%), and alanine (Ala) (3.31% and 3.30%), but low in cysteine (Cys) (0.05% and 0.03%), methionine (Met) (0.70% and 0.44%), and threonine (Thr) (0.78% and 0.69%). Due to the treatment of acid hydrolysis, the tryptophan was failed to detect. The total amino acid (TAA) of protein and polypeptide was accounted for nearly same proportion, with 40.95% and 38.86% dry cell weight (DCW), respectively, indicating that hydrolyzed WBY protein by trypsin had better retention of amino acid diversity, although a small part of amino acids was degraded in weak alkaline environment (pH 7.8). Furthermore, the EAA/TAA and EAA/NEAA of protein and polypeptide of WBY closed to but below FAO/WHO ideal protein model, which partly because the tryptophan—one of the EAA—was undetected. Furthermore, the hydrophobic amino acid (HAA) of protein and polypeptide was 16.32 g/100 g and 15.81 g/100 g, respectively, indicating that the protein and polypeptide of WBY manifested good solubility in water-lipid solution and better ability to interact with free radicals [29]. Yuan et al., reported that the polypeptide extracted from *Zanthoxylum bungeanum* Maxim seeds (ZBMS) has 14.90% HAA and exhibited great antioxidant activity [30]. The flavor amino acids (FAA) of protein and polypeptide was 8.83 g/100 g and 8.09 g/100 g, respectively; thus, WBY protein as well as enzyme-hydrolyzed products presented umami flavor. It was remarkable that the arginine (Arg) is a great energy source for the growth of bacteria [31], and that of protein and polypeptide was 1.99 g/100 g and 2.02 g/100 g, respectively.

As shown in Figure 5B, the AAS of AAA (phenylalanine + tryptophane) and Lys were the highest in WBY protein and polypeptide, accounting for 106.05%, 87.11% and 104.74%, 71.33%, respectively. The content of AAA is a bit higher than Food and Agricultural Organization (FAO) AA requirements for adults. The score of leucine (Leu) and valine (Val) in WBY protein and polypeptide was slightly above 50%, and that of isoleucine (Ile) was 48.33% and 46.67% for WBY protein and polypeptide, respectively. In addition, the threonine (Thr) and SAA (methionine + cysteine) were the limiting amino acid for their lower AAS.

Overall, the results of composition and AAS revealed that WBY protein and polypeptide were better resources to provide abundant and diverse amino acids, such as Glu, Gly, Lys, Ala, and Leu. Moreover, the amino acid contents of Lys, Phe, and Met in WBY protein and polypeptide were significantly higher than that in grains [32], thus the insufficient amino acids of grains can be replenished by WBY protein and polypeptide.

### 2.5. Antioxidant Activity

#### 2.5.1. Hydroxyl Radical Scavenging Assay

Figure 6A depicts the hydroxyl radical scavenging ability of vitamin C (Vc), protein, and polypeptide. When the concentration of Vc and polypeptide was maximum (0.5 mg/mL and 35 mg/mL), the antioxidant abilities of both samples were nearly same, at approximately 98.76% and 95.10%, respectively. The scavenging rate of WBY protein increased from 80.22% to 88.41% with the concentration increased from 15 mg/mL to 35 mg/mL, which demonstrated that WBY protein had scavenging ability against hydroxyl radical but not as good as WBY polypeptide. Furthermore, there was no significant difference was observed between Vc and polypeptide on scavenging rate even at small concentration (0.1 mg/mL and 15 mg/mL), indicating that WBY polypeptide had great hydroxyl radical scavenging activity and almost equaled to Vc. The results are lined with the report that the peptides produced by hydrolyzing whey protein with trypsin showed strong scavenging activity of O_2_^−^^⋅^ and ^⋅^OH and cytoprotective effect [28]. 

#### 2.5.2. DPPH Radical Scavenging Assay

As shown in Figure 6B, the scavenging activity of Vc, protein, and polypeptide against DPPH radical increased in a dose-dependent manner. The maximum scavenging rate of Vc, protein and polypeptide were 99.05%, 83.41% and 98.37%, respectively, at maximum concentration. The scavenging activity of WBY polypeptide was 1.17-fold that of WBY protein when the sample concentration was 15 mg/mL. Furthermore, the DPPH radical scavenging rate of WBY polypeptide and Vc increased almost same but still higher than WBY protein, which manifested that the WBY polypeptide has great scavenging activity against DPPH radical. The scavenging rate of ZBMS polypeptide was 78.45% at a sample concentration of 20 mg/mL [30], which indicated that the higher scavenging activity against DPPH was related to higher polypeptide concentration [33].

#### 2.5.3. ABTS Radical Scavenging Assay

The scavenging rate of ABTS radical for WBY polypeptide was increased from 53.37% to 69.41%, and that of WBY protein increased from 45.22% to 64.33%, as the sample concentration increased from 15 mg/mL to 35 mg/mL. Although the antioxidant activity of WBY polypeptide was slightly higher than WBY protein on scavenging ABTS, WBY polypeptide has a weaker scavenging activity than Vc. The scavenging ability on ABTS radical is related to various factors, such as the structure of polypeptide, the composition of AA, the ways and degree of hydrolysis [21].

Moreover, as shown in Table 7, WBY polypeptide exhibited scavenging activity of three radicals with IC_50_ values varied from 2.68 to 12.86 mg/mL, which was significantly lower than the IC_50_ values of WBY protein except hydroxyl radical (*p* < 0.05). Both WBY polypeptide and protein possessed stronger scavenging activity against hydroxyl radical with no significant difference in IC_50_ values (*p* > 0.05). Overall, the results of scavenging rate and IC_50_ revealed that the polypeptide extracted from WBY had greater antioxidant activity compared with protein. 

## 3. Materials and Methods

### 3.1. Materials

WBY paste was provided by Tsingtao Brewery Co., Ltd. (Huangshi, Hubei). BCA protein concentration determination kit, SDS-PAGE gel quick preparation kit, SDS-PAGE electrophoresis buffer, coomassie brilliant blue R–250 and 5× SDS-PAGE sample loading buffer were purchased from Beyotime Institute of Biotechnology Co., Ltd. M5 Prestained Plus Protein Ladder (10–250 kDa) with 40 kD was obtained from Mei5 Biotechology. CO., Ltd. (Beijing, China). Trypsin (1:250) and alkaline protease (1:20,000) were purchased from Saiguo biotech CO., Ltd. (Guangzhou, China) and Solarbio Science and Technology CO., Ltd. (Beijing, China), respectively. Methanol acid, sulfuric acid, hydrochloric acid, glacial acetic acid, ethanol, ferrous sulfate, hydrogen peroxide, salicylic acid, DPPH, ABTS, potassium persulfate, PBS (pH 7.4), disodium edetate, and pyrogallic acid were obtained from Sinopharm Chemical Reagent Co., Ltd. (Shanghai, China). All chemicals used in this study were of analytical grade (AR) or guaranteed grade (GR).

### 3.2. Process Optimization of Extracting Protein

#### 3.2.1. Pretreatment

WBY paste was centrifuged for 20 min at 6000 rpm to collect the beer yeast precipitate, then the precipitate was washed three times by distilled water. After that, the collected WBY was pre-frozen at −80 °C for 48 h in a Thermo 900 Series freezer (Shanghai, China) and dried by an Alpha 2–4 Freeze Dryer (Christ, Germany) at −80 °C for 72 h at 0.05 mbar. The dried WBY was ground to powder and stored in amber flasks at 4 °C until use.

#### 3.2.2. Ultrasonic-Assisted Extraction

The protein of dried WBY powder was extracted by ultrasonic-assisted extraction under ice-bath with distilled water as the extracting agent. The ultrasonic cell disruptor JY92-IIN (Ningbo, China) was employed to break up yeast cells. After ultrasonic processing, the mixture was centrifuged for 20 min at 6000 rpm to collect the supernatant containing the crude protein. The protein was precipitated by 60% (*v*/*v*) ammonium sulfate under ice bath for 15 min and centrifuged by 10,000 g for 10 min to remove supernatant. After re-dissolved in distilled water, the solution was desalted using a 10 kDa cut-off ultrafiltration membrane (Milipore, Burlington, MA, USA) and freeze dried to solid powder using an Alpha 2–4 Freeze Dryer (Christ, Germany) at −80 °C for 72 h at 0.05 mbar.

#### 3.2.3. Extraction Rate and Purity

The total content of crude protein in WBY was quantified using Kjeldahl method described by [34]. Furthermore, the content of water-soluble protein in supernatant of WBY extraction was measured using BCA protein concentration determination kit, according to the manufacturer’s instructions. First, the standard curve of bovine serum albumin (BSA) was acquired by measuring the absorbance value of different concentrations of BSA standard solution at 562 nm after reacted with working liquid of BCA. Moreover, the absorbance value of supernatant of extraction after treatment was measured under the same wavelength, and the content of water-soluble protein in supernatant was calculated using the standard curve.

The extraction rate of WBY protein was calculated as follows:$$\mathrm{Extraction}\;\mathrm{rate}\;\left(\%\right)={\;C}_{2}\;/{\;C}_{1}\times 100$$
where C_1_ refers to the total content of crude protein in WBY (g); C_2_ refers to the content of wter-soluble protein in supernatant (g).

The purity of WBY protein was calculated as follows [35]:$$\mathrm{Purity}\;\left(\%\right)={\;m}_{2}\;/{\;m}_{1}\times 100$$
where m_1_ refers to the total mass of WBY protein powder (g); m_2_ refers to the mass of protein in powder (g).

#### 3.2.4. Single-Factor Experimental Design

It has been reported that the yield of ultrasonic-assisted extraction is influenced by several factors, such as pH, power, solid-liquid ratio, time and temperature [36]. Based on our preliminary experiment, the following three factors were investigated by a single-factor experiment, with other conditions unchanged: the power of ultrasound (250, 300, 350, 400, and 450 W), the pH of extraction (5.5, 6.5, 7.5, 8.5, and 9.5), and the solid-liquid ratio (6%, 8%, 10%, 12%, and 14%). The pH of extraction was adjusted by 0.1 mol/L Na_2_CO_3_ or HCl. Each ultrasonic assay was continued for 12 min with the intermittent ratio of 1:1.

#### 3.2.5. Experimental Design of Response Surface Method (RSM)

RSM with a 3-factor-3-level Box–Behnken design was employed to analysis the influence of independent variables during ultrasonic-assisted extraction in order to obtain the maximum extraction rate of WBY protein. The independent variables were power of ultrasound, pH of extraction and solid-liquid ratio. A second-order polynomial equation was used for predicting Y, as follows: $$Y=b_{0}+\overset{3}{\underset{i=1}{\sum }}b_{i}x_{i}+\overset{3}{\underset{i=1}{\sum }}b_{\mathrm{ii}}x_{i}^{2}+\overset{2}{\underset{i=1}{\sum }}\overset{3}{\underset{j=i+1}{\sum }}b_{\mathrm{ij}}x_{i}x_{j}$$
where Y is the response for the extraction rate of WBY protein; b_0_, b_j_, b_ii_, and b_ij_ are the constant coefficients of intercept, linear, quadratic, and interaction terms, respectively; and X_i_ and X_j_ are uncoded independent variables.

### 3.3. Process Optimization of Preparing Polypeptide

#### 3.3.1. Enzymatic Hydrolysis and Degree of Hydrolysis (DH)

Trypsin and alkaline protease were used for hydrolyzing WBY protein to obtain WBY polypeptide. The supernatant of WBY extraction under optimal conditions was added with the 0.1% trypsin (250 U, pH 8.5, 37 °C) and 0.125% alkaline protease (250 U, pH 9, 45 °C), respectively, and incubated for 24 h with other conditions kept unchanged. The protein in the mixture was precipitated by added 10% (V/V) trichloroacetic acid (TCA) then removed by centrifuging for 5 min at 5000 rpm. The crude polypeptide solution was purified by a 10 kDa cut-off ultrafiltration membrane and freeze dried to powder. The content of polypeptide in supernatant was determined every 4 h by Kjeldahl method after inactivated enzymes [27,28]. 

The DH estimated based on the method described by [27]:$$\mathrm{DH}\;\left(\%\right)=\left({\mathrm{PS}}_{1}-{\;\mathrm{PS}}_{0}\right)\;/{\;P}_{\mathrm{total}}\times 100$$
where PS_0_ refers to the content of soluble protein after precipitated by 10% TCA before adding of enzyme(mg/mL); PS_1_ refers to the content of soluble protein after precipitated by 10% TCA after adding of enzyme (mg/mL); P_total_ refers to the total content of soluble protein (mg/mL).

#### 3.3.2. Extraction Rate and Purity

The extraction rate of WBY polypeptide was calculated as follows:$$\mathrm{Extraction}\;\mathrm{rate}\;\left(\%\right)={\;C}_{3}\;/{\;C}_{2}\times 100$$
where C_3_ refers to the content of water-soluble nitrogen in supernatant after removed protein (g). 

The purity of WBY polypeptide was measured as same as that of protein.

#### 3.3.3. Single-Factor Experimental Design

It has been shown that the hydrolysis of protein is influenced by several factors, such as temperature, pH, enzyme concentration, and time. Based on our preliminary experiment, the following three factors were investigated by a single-factor experiment: the temperature (30, 35, 40, 45, and 50 °C), the pH of extraction (7, 7.5, 8, 8.5, and 9), and the enzyme concentration (0.5%, 1.0%, 1.5%, 2.0%, and 2.5%). The time of enzymatic hydrolysis was 12 h with other conditions unchanged, and the enzymes were inactivated by heating at 100 °C for 10 min.

#### 3.3.4. RSM Experimental Design

The RSM experimental design was the same as 3.2.5 with Y is the response for the extraction rate of WBY polypeptide. The independent variables were temperature (35–45 °C), pH of extraction (7–8), and enzyme concentration (0.5–1.5%) based on the results of single-factor experiments.

### 3.4. Molecular Weight

SDS-polyacrylamide gel electrophoresis (SDS-PAGE) was used for measuring the molecular weight of protein and polypeptide of WBY using the methods described by [28]. 8% separating gel and 5% stacking gel were obtained using SDS-PAGE gel quick preparation kit. After electrophoresis for 90 min with 15 mA and 80 V, the gel was dyed an hour with dyeing solution and decolorized one night with decoloring solution, the gel image was obtained by Bio-RAD ChemiDocXRS (Hercules, CA, USA) subsequently.

### 3.5. Amino Acid Composition

The amino acid composition of protein and polypeptide powder of WBY were determined using Hitachi l-8900 type automatic amino acid analyzer (Tokyo, Japan) [29].

The amino acid score (AAS) was calculated as follows:$$\mathrm{AAS}=\frac{A_{1}}{A_{2}}\times 100$$
where A_1_ is the amino acid in sample (mg/g); A_2_ is the amino acid in FAO/WHO model (mg/g).

### 3.6. Antioxidant Activity

#### 3.6.1. Hydroxyl Radical Scavenging Assay

The extracted solution of protein and polypeptide of WBY was diluted to the same concentration (15, 20, 25, 30, and 35 mg/mL) with distilled water, respectively. 1 mL of FeSO_4_ solution (9 mmol/L), salicylic acid ethanol-water solution (9 mmol/L) and protein and polypeptide of WBY (15, 20, 25, 30, and 35 mg/mL) were mixed, respectively, and stood for 10 min. Meanwhile, the same volume of FeSO_4_ solution, salicylic acid ethanol-water solution and Vc solution (0.1, 0.2, 0.3, 0.4, and 0.5 mg/mL) was mixed as positive control. Next, 1 mL of H_2_O_2_ solution (8.8 mol/L) was added to the mixture and left to stand for 30 min at 37 °C. The groups where WBY protein and polypeptide were replaced by distilled water were viewed as blank, and the groups in which H_2_O_2_ solution was replaced by distilled water were considered as control. The absorbance values were measured using a UV-4802 spectrophotometer (Wuhan, China) with distilled water as blank. The hydroxyl radical scavenging rate was calculated based on the absorbance values measured at 510 nm:$$\mathrm{Scavenging}\;\mathrm{rate}\;\left(\%\right)=\left[1-\left(A_{2}-{\;A}_{1}\right)\;/{\;A}_{0}\right]\times 100$$
where A_0_ refers to the absorbance of blank groups, A_1_ is the absorbance of control groups, and A_2_ refers to the absorbance of protein samples, polypeptide, and Vc samples.

#### 3.6.2. DPPH Radical Scavenging Assay

1 mL of 0.1 mM DPPH ethanol-water solution was mixed with 1 mL of WBY protein and polypeptide (15, 20, 25, 30, and 35 mg/mL), respectively, and reacted in the dark for 30 min. The same volume of DPPH solution and Vc solution (0.1, 0.2, 0.3, 0.4, and 0.5 mg/mL) was mixed as positive control. At same time, the groups where WBY protein and polypeptide were replaced by distilled water were viewed as blank, and the groups where DPPH solution was replaced by absolute ethanol were considered as the control. The DPPH scavenging rate was calculated based on the absorbance values measured at 517 nm:$$\mathrm{Scavenging}\;\mathrm{rate}\;\left(\%\right)=\left[1-\left(A_{2}-{\;A}_{1}\right)\;/{\;A}_{0}\right]\times 100$$
where A_0_ refers to the absorbance of blank groups, A_1_ refers to the absorbance of control groups, and A_2_ refers to the absorbance of protein samples, polypeptide, and Vc samples.

#### 3.6.3. ABTS Scavenging Assay

The ABTS scavenging activity of WBY protein and polypeptide was measured using the methods depicted by María et al. with little modification [28]. The ABTS stock solution was prepared by mixing 1 mL ABTS solution (7.4 mmol/L) and 1 mL potassium persulfate solution (2.6 mmol/L) and reacting at dark for 12 h. Hereafter, 200 μL ABTS stock solution which was diluted using phosphate buffer (the absorbance value closed to 0.7) was mixed with 10 μL WBY protein and polypeptide samples. The groups that replaced WBY protein and polypeptide with Vc solution, with other conditions unchanged, were considered as positive control. After reacted at dark for 6 min, the final absorbance value was measured at 735 nm and the ABTS scavenging rate was calculated as follows:$$\mathrm{Scavenging}\;\mathrm{rate}\;\left(\%\right)=(1-\frac{A_{1}\;}{A_{0}})\times 100$$
where A_0_ refers to the absorbance of diluted ABTS stock solution, A_1_ refers to the absorbance of protein samples, polypeptide, and Vc samples.

#### 3.6.4. IC_50_ Values

The values of concentrations of WBY protein and polypeptide were changed into logarithmic, and the probit model of logarithmic and IR was obtained using SPSS software 19. The estimate concentration that corresponding to probability of 0.5 was IC_50_.

### 3.7. Statistical Analyses

All experiments were performed in triplicate and data were expressed as mean values ± standard deviations in this study, using the software Origin of version 2018 (Originlab Corporation, Northampton, MA, USA). The RSM experimental design of extracting protein and polypeptide from WBY was implemented using Design-Expert of version 11 (StatEase Inc., Minneapolis, MN, USA); the independent samples *t*-test and one-way analysis of variance (ANOVA) with Tukey’s posttest at 95% probability were used for identifying differences among different groups. *p* < 0.05 was viewed as statistically significant. All calculations were performed using the software Mathematica of version 8 (Wolfram Research Inc., Champaign, IL, USA).

## 4. Conclusions

In summary, the water-soluble protein and polypeptide of WBY were extracted by ultrasonic-assisted extraction and enzymatic hydrolysis, respectively, which yielded 73.94% of protein and 61.24% of polypeptide under optimal condition. Moreover, the SDS-PAGE gel electrophoresis showed that the bands of WBY protein ranged from 25 to 50 kDa, and that of WBY polypeptide were located blew 10 kDa because of hydrolyzing by 1.0% (*v*/*v*) trypsin. The results of amino acid composition showed that the protein and polypeptide were composed of 17 amino acids, and the TAA was accounted for 40.95% and 38.86% DCW, respectively. Furthermore, Lys and AAA closed to FAO/WHO ideal protein model and the Thr and SAA were the limiting amino acids. Furthermore, the results of scavenging rate and IC_50_ revealed that the polypeptide extracted from WBY has greater antioxidant activity compared with protein. When WBY polypeptide concentration reached 35 mg/mL, the scavenging activity of polypeptide against hydroxyl radical, DPPH radical, and ABTS radical reached 95.10%, 98.37%, and 69.41%, respectively. Overall, the protein and polypeptide of WBY with abundant amino acid can be efficiently extracted using ultrasonic-assisted extraction and trypsin-hydrolysis, and the WBY polypeptide with small molecules weight possessed greater antioxidant activity when evaluated by hydroxyl radical, DPPH radical, and ABTS radical. These findings are significant for making full use of WBY through green and efficient extraction method, which indicating that protein and polypeptide of WBY have a promising application in the food, feed, and pharmaceutical industry.

## Figures and Tables

**Figure 1 molecules-27-06825-f001:**
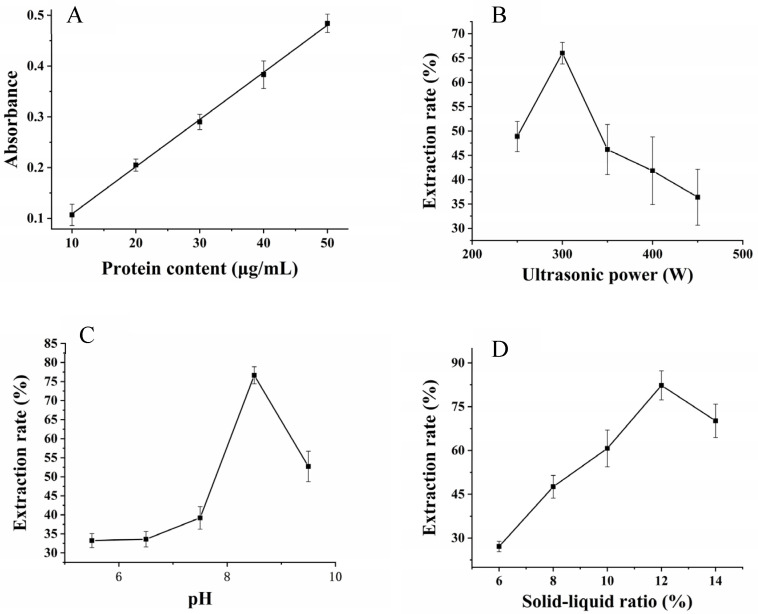
The standard curve of protein (**A**) and single factor experiments for power of ultrasound (**B**), pH of extraction (**C**) and solid-liquid ratio (**D**).

**Figure 2 molecules-27-06825-f002:**
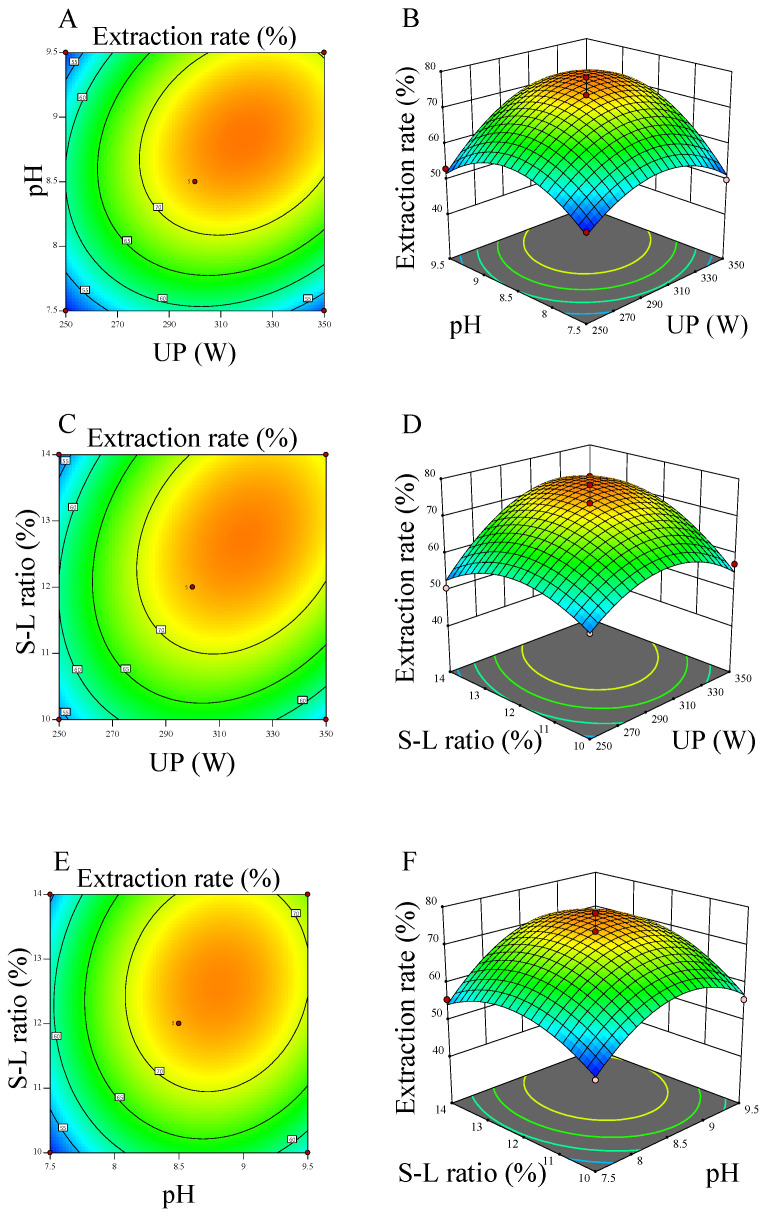
Response surface plots showing the effects of extraction pH, UP (ultrasound power) and solid-liquid (S-L) ratio on the WBY protein extraction. (**A**,**B**) effect of interaction between extraction pH and UP; (**C**,**D**) effect of interaction between UP and S-L ratio; (**E**,**F**) effect of interaction between the extraction pH and S-L ratio.

**Figure 3 molecules-27-06825-f003:**
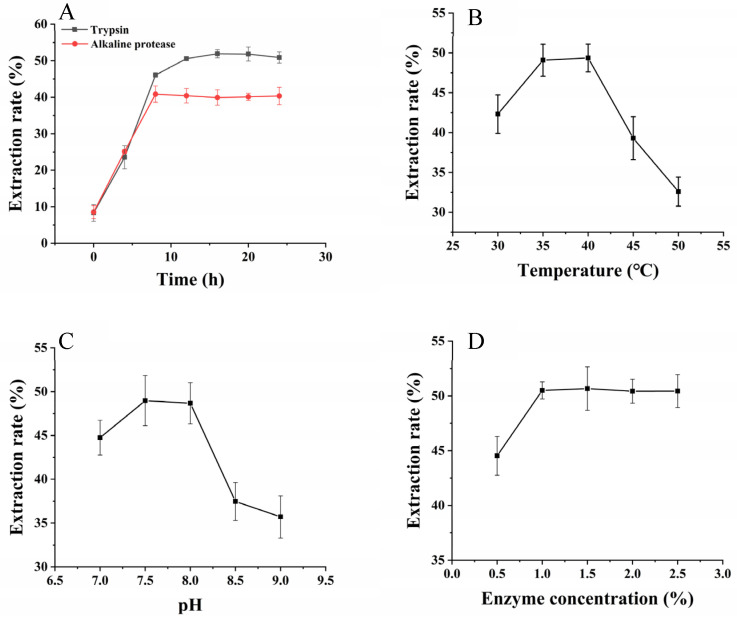
Comparison of two kinds of enzymes (**A**) and single-factor experiments of reaction temperature (**B**), reaction pH (**C**) and enzyme concentration (**D**).

**Figure 4 molecules-27-06825-f004:**
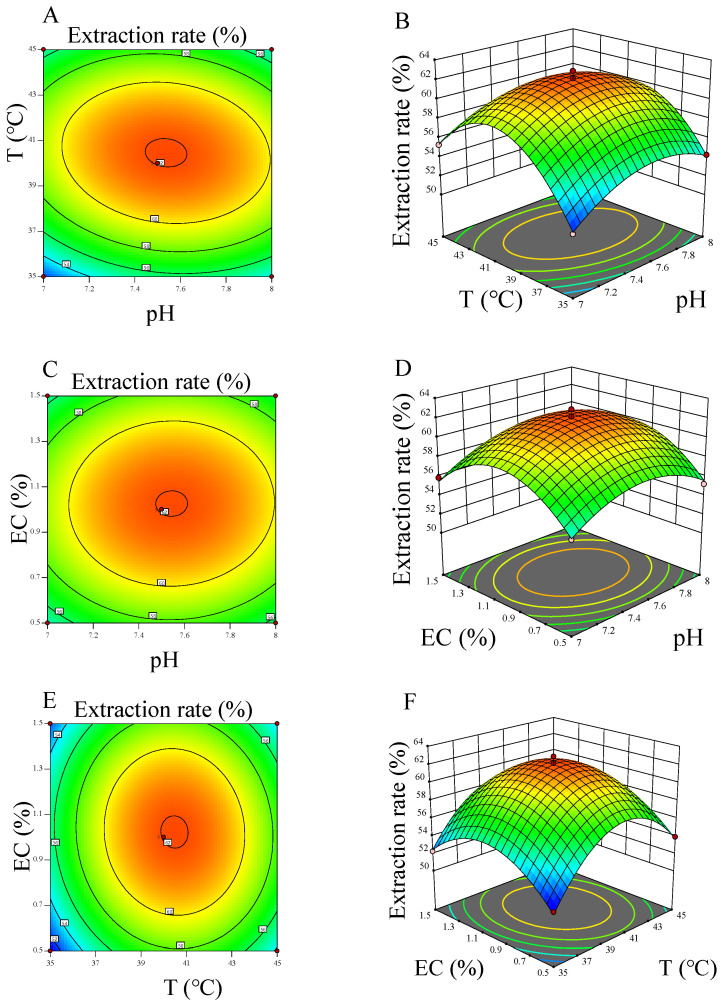
Response surface plots showing the effects of T (temperature), extraction pH, EC (enzyme concentration) on the WBY polypeptide extraction. (**A**,**B**) effect of interaction between T and extraction pH; (**C**,**D**) effect of interaction between EC and extraction pH; (**E**,**F**) effect of interaction between T and EC.

**Figure 5 molecules-27-06825-f005:**
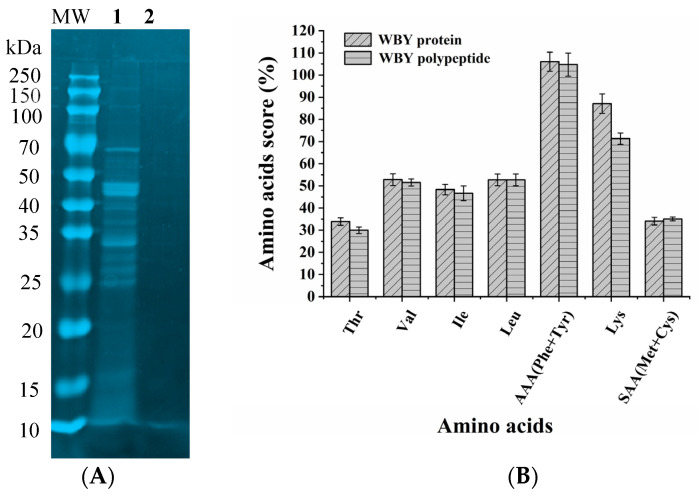
Coomassie-stained SDS-PAGE electrophoretic profile (**A**) of WBY protein (**1**) and polypeptide (**2**) and amino acids score of WBY protein and polypeptide (**B**). M5 Prestained Plus Protein Ladder (10–250 kDa) with 40 kDa as the protein weight standard.

**Figure 6 molecules-27-06825-f006:**
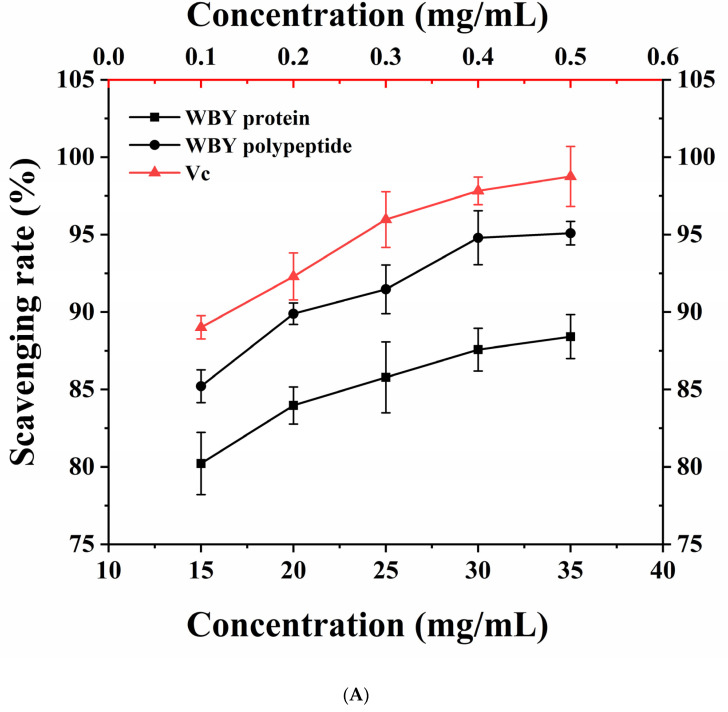
Antioxidant activity against hydroxyl radical (**A**), DPPH radical (**B**) and ABTS radical (**C**).

**Table 1 molecules-27-06825-t001:** Extracting protein and polypeptide from WBY.

	Methods	Yield/Content	DH	Purity	Antioxidant Activity			
					⋅OH	DPPH⋅	O_2_^−^⋅	ABTS^+^⋅
Protein	Bienzymatic hydrolysis [21]	29.67% (Y)	28.88%	NR	NR			1.59 gTEAC
	Hydrothermal decomposition (250 °C, 20 min) [23]	78% (Y)	NR	NR	NR			NR
	Autolysis (70 °C, 4 h) [24]	49% (C)	NR	NR	NR			NR
	Antolysis (45 °C, 72 h) [25]	76.4% (Y)	NR	NR	NR			NR
Peptide	Alcalase hydrolysis [26]	NR	NR	NR	IC50 < 0.8 mg/mL			NR
	CCE hydrolysis [24]	NR	NR	30–69%	NR			NR

CCE-*C. cardunculus* extraction; DH-degree of hydrolysis; TEAC-Trolox equivalent antioxidant capacity; NR-Not report.

**Table 2 molecules-27-06825-t002:** Response surface test design and WBY protein extraction rate.

Run	X_1_	X_2_	X_3_	Extraction Rate%
1	250	7.5	12	50.92
2	350	7.5	12	49.93
3	250	9.5	12	53.16
4	350	9.5	12	68.84
5	250	8.5	10	53.75
6	350	8.5	10	57.23
7	250	8.5	14	50.63
8	350	8.5	14	70.71
9	300	7.5	10	49.86
10	300	9.5	10	55.64
11	300	7.5	14	55.62
12	300	9.5	14	68.55
13	300	8.5	12	78.48
14	300	8.5	12	70.40
15	300	8.5	12	72.92
16	300	8.5	12	73.66
17	300	8.5	12	71.54

**Table 3 molecules-27-06825-t003:** Variance analysis of WBY protein extraction rate.

Source of Variance	Sum of Squares	Free Degree	Mean Square Deviation	F Value	*p* Value	Significance
Model	1591.21	9	176.80	21.67	0.0003	**
X_1_	183.36	1	183.36	22.47	0.0021	**
X_2_	198.01	1	198.01	24.26	0.0017	**
X_3_	105.85	1	105.85	12.97	0.0087	**
X_1_X_2_	69.72	1	69.72	8.54	0.0222	*
X_1_X_3_	68.89	1	68.89	8.44	0.0228	*
X_2_X_3_	12.60	1	12.60	1.54	0.2540	N
X_1_^2^	304.57	1	304.57	37.32	0.0005	**
X_2_^2^	354.83	1	354.83	43.48	0.0003	**
X_3_^2^	194.98	1	194.98	23.89	0.0018	**
Residual	57.12	7	8.16			
Lack of Fit	19.23	3	6.41	0.6767	0.6103	N
Pure Error	37.89	4	9.47			
Cor Total	1648.34	16				
R^2^	0.9653					

**-Very significant difference (*p* < 0.01); *-Significant difference (*p* < 0.05); N-Insignificant difference (*p* > 0.05).

**Table 4 molecules-27-06825-t004:** Response surface test design and WBY polypeptide extraction rate.

Run	X_1_	X_2_	X_3_	Extraction Rate%
1	7	35	1	51.68
2	8	35	1	54.26
3	7	45	1	55.31
4	8	45	1	55.03
5	7	40	0.5	54.79
6	8	40	0.5	55.21
7	7	40	1.5	55.87
8	8	40	1.5	56.65
9	7.5	35	0.5	51.2
10	7.5	45	0.5	53.93
11	7.5	35	1.5	52.26
12	7.5	45	1.5	53.51
13	7.5	40	1	61.92
14	7.5	40	1	62.15
15	7.5	40	1	61.48
16	7.5	40	1	61.67
17	7.5	40	1	62.88

**Table 5 molecules-27-06825-t005:** Variance analysis of WBY polypeptide extraction rate.

Source of Variance	Sum of Squares	Free Degree	Mean Square Deviation	F Value	*p* Value	Significance
Model	250.16	9	27.80	108.35	< 0.0001	**
X_1_	1.53	1	1.53	5.97	0.0446	*
X_2_	8.78	1	8.78	34.22	0.0006	**
X_3_	1.25	1	1.25	4.87	0.0632	N
X_1_X_2_	2.04	1	2.04	7.97	0.0256	*
X_1_X_3_	0.0324	1	0.0324	0.1263	0.7328	N
X_2_X_3_	0.5476	1	0.5476	2.13	0.1874	N
X_1_^2^	26.79	1	26.79	104.44	< 0.0001	**
X_2_^2^	124.03	1	124.03	483.50	< 0.0001	**
X_3_^2^	62.98	1	62.98	245.51	< 0.0001	**
Residual	1.80	7	0.2565			
Lack of Fit	0.6151	3	0.2050	0.6947	0.6018	N
Pure Error	1.18	4	0.2952			
Cor Total	251.95	16				
R^2^	0.9929					

**-Very significant difference (*p* < 0.01); *-Significant difference (*p* < 0.05); N-Insignificant difference (*p* > 0.05).

**Table 6 molecules-27-06825-t006:** Amino acid composition of protein and polypeptide of WBY.

Amino Acids		Protein (%)	Polypeptide (%)
EAA	Thr	0.78 ± 0.14	0.69 ± 0.03
	Val ^#^	2.06 ± 0.06	2.01 ± 0.12
	Ile ^#^	1.45 ± 0.11	1.40 ± 0.03
	Leu ^#^	3.11 ± 0.09	3.11 ± 0.14
	Phe ^#,a^	1.79 ± 0.05	1.77 ± 0.05
	Lys ^*^	3.92 ± 0.10	3.21 ± 0.02
	Met ^#^	0.70 ± 0.03	0.44 ± 0.03
NEAA	Asp *^,b^	2.37 ± 0.04	1.96 ± 0.01
	Ser	1.82 ± 0.06	1.01 ± 0.09
	Glu *^,b^	6.46 ± 0.15	6.13 ± 0.13
	Gly	5.26 ± 0.02	6.02 ± 0.15
	Ala ^#^	3.31 ± 0.12	3.30 ± 0.10
	Tyr ^#,a^	2.24 ± 0.08	2.21 ± 0.06
	His *	2.03 ± 0.02	2.01 ± 0.13
	Arg *	1.99 ± 0.07	2.02 ± 0.07
	Pro ^#^	1.61 ± 0.11	1.54 ± 0.11
	Cys ^#^	0.05 ± 0.02	0.03 ± 0.01
	EAA	13.81 ± 0.47	12.63 ± 0.34
	NEAA	27.14 ± 0.56	26.23 ± 0.70
	TAA	40.95 ± 1.04	38.86 ± 1.05
	HAA	16.32 ± 0.55	15.81 ± 0.53
	CAA	16.77 ± 0.31	15.33 ± 0.29
	AAA	4.03 ± 0.11	3.98 ± 0.09
	FAA	8.83 ± 0.16	8.09 ± 0.11

EAA-Essential amino acid; NEAA-Non-essential amino acids; TAA-Total amino acids; HAA-Hydrophobic amino acid (#); CAA-Charged amino acids (*); AAA-Aromatic amino acids (a); FAA-Flavor amino acids (b).

**Table 7 molecules-27-06825-t007:** IC_50_ values of WBY protein, WBY polypeptide and Vc.

Samples	Hydroxyl Radical	DPPH Radical	ABTS Radical
	IC50 (mg/mL)	IC50 (mg/mL)	IC50 (mg/mL)
protein	2.83 ± 0.13 ^b^	8.67 ± 0.15 ^c^	18.75 ± 0.35 ^c^
polypeptide	2.68 ± 0.11 ^b^	5.63 ± 0.31 ^b^	12.86 ± 0.26 ^b^
Vc	0.018 ± 0.08 ^a^	0.015 ± 0.06 ^a^	0.038 ± 0.11 ^a^

Values (mean ± standard deviation of triplicate) in the same column with different superscripts are significantly different (*p* < 0.05).

## Data Availability

The data are contained within the article.

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
