# Peer review of "Process Optimization, Amino Acid Composition, and Antioxidant Activities of Protein and Polypeptide Extracted from Waste Beer Yeast"

_molecules, 2022, doi:10.3390/molecules27206825_

Round 1
Reviewer 1 Report
1.There is already some literature on protein extraction and peptide preparation from waste beer yeast, which should appear in the introduction。
2.How is the single factor determined when extracting protein? Why not consider the extraction time?
3.How is a protein or peptide prepared for analysis of amino acid composition and antioxidant activity? Especially when analyzing amino acids, do you use solid powder? How is it prepared? What is the purity?
4.Why was the degree of proteolysis not measured when polypeptide was extracted? Polypeptides are derived from proteolysis, with high degree of hydrolysis and high polypeptide yield
5When extracting polypeptide, the optimal conditions for the regression equation are extraction pH 7.54, the temperature 40.5℃, and enzyme concentration was 1.0%. Why is there no rounding during verification? 7.54 and 40.5°C are difficult to precisely control
6.This part of extraction lacks necessary discussion. Whether it is protein extraction or polypeptides extraction, it should be compared with the current literature, such as how the extraction rate of other methods is compared with this paper.
7.The conclusion is too long and needs to be refined.
Reviewer 2 Report
The paper entitled ‘’Process optimization, amino acid composition, and antioxidant activities of protein and polypeptide extracted from waste beer yeast’’ treats an interesting topic related to repurposing of the waste beer yeast.
There are some adjustments that need to be made:
1. The acronym ‘’Vc’’ must be explained at its first use.
2. The coefficient of determination (R2) in the models of extraction rate should be useful for underlying the accuracy of the experimental data.
3. The Discussion part by reporting to literature is relative scarce when the results of the process optimization of extracting protein and polypeptide respectively are presented.
4. Some figures (i.e. Fig. 2B, 2D, 3C) are relative unclear from graphical point of view.
5. The References should be written in agreement with the MDPI requirements.
Reviewer 3 Report
Manuscript molecules-1908736 is very well written; clear, precise, and easy to understand. However, some comments should be addresed before their consideration for publication in Molecules journal.
Why authors use % of radical inhibition instead IC50 (that indicates how much of a particular inhibitory susbtance (e.g. protein/peptides) is needed to inhibit, in vitro, a given biological process or biological component by 50%), which is standarized by the protein/peptide concentration and allows to compare values from others investigation. I observed that in the Figure 6 is reported the antixoidant acvitiy in terms of % of radical inhibition at different concentrarions, thus, it is feasible to calculate IC50 values.
At the Line 198-199 the authors mentioned that the experimental value obtained with optimal parameters is closed to the predicted value. However, these values were compared statistically? Please add specify and done the pertinent statistical test.
